# Spectrum of High-Risk Mutations among Breast Cancer Patients Referred for Multigene Panel Testing in a Romanian Population

**DOI:** 10.3390/cancers15061895

**Published:** 2023-03-22

**Authors:** Iulian Gabriel Goidescu, Georgiana Nemeti, Mihai Surcel, Gabriela Caracostea, Andreea Roxana Florian, Gheorghe Cruciat, Adelina Staicu, Daniel Muresan, Cerasela Goidescu, Roxana Pintican, Dan Tudor Eniu

**Affiliations:** 1Obstetrics and Gynecology I, Mother and Child Department, University of Medicine and Pharmacy “Iuliu Hatieganu”, 400006 Cluj-Napoca, Romania; 2Department of Internal Medicine, Medical Clinic I—Internal Medicine, Cardiology and Gastroenterology, University of Medicine and Pharmacy “Iuliu Hatieganu”, 400006 Cluj-Napoca, Romania; 3Department of Radiology, Iuliu Hatieganu University of Medicine and Pharmacy, 400347 Cluj-Napoca, Romania; 4Department of Surgery 2, University Emergency Hospital, University of Medicine and Pharmacy “Iuliu Hatieganu”, 400006 Cluj-Napoca, Romania

**Keywords:** BRCA1/2, TP53, PALB2, hereditary breast cancer, Li Fraumeni syndrome

## Abstract

**Simple Summary:**

Multigene panel testing for Hereditary Breast and Ovarian Cancer using next generation sequencing is the new standard for the identification of individuals with cancer predisposing gene variants. The purpose of our research was to investigate the genetic variants implicated in hereditary breast cancer predisposition in 255 Romanian individuals referred for management. In our cohort, gene analysis found most oncogenic mutations in BRCA 1/2 genes, followed by the high penetrance PALB2 and TP53 genes, while in the CDH1 and STK11 genes only VUS mutations were identified. Multigene testing of populations of different geographical and ethnical backgrounds, followed by data merging, will enable us to map the clinical significance of genetic variants and tailor management decisions for Hereditary Breast and Ovarian Cancer patients.

**Abstract:**

(1) Background: Multigene panel testing for Hereditary Breast and Ovarian Cancer (HBOC) using next generation sequencing (NGS) is becoming a standard in medical care. There are insufficient genetic studies reported on breast cancer (BC) patients from Romania and most of them are focused only on BRCA 1/2 genes (Breast cancer 1/2). (2) Methods: NGS was performed in 255 consecutive cases of BC referred for management in our clinic between 2015–2019. (3) Results: From the 171 mutations identified, 85 were in the high-penetrance BC susceptibility genes category, 72 were pathogenic genes, and 13 genes were in the (variants of uncertain significance) VUS genes category. Almost half of the mutations were in the BRCA 1 gene. The most frequent BRCA1 variant was c.3607C>T (14 cases), followed by c.5266dupC (11 cases). Regarding BRCA-2 mutations we identified c.9371A>T (nine cases), followed by c.8755-1G>A in three cases, and we diagnosed VUS mutations in three cases. We also identified six pathogenic variants in the PALB2 gene and two pathogenic variants in (tumor protein P 53) TP53. (4) Conclusions: The majority of pathogenic mutations in the Romanian population with BC were in the BRCA 1/ 2 genes, followed by PALB2 (partner and localizer of BRCA2) and TP53, while in the CDH1 (cadherin 1) and STK11 (Serine/Threonine-Protein Kinase) genes we only identified VUS mutations.

## 1. Introduction

Hereditary Breast and Ovarian Cancer Syndrome (HBOC) is an autosomal dominant inherited condition caused by the presence of pathogenic variants in the BRCA1 and BRCA2 genes [1]. Until recently, genetic diagnosis of this condition was mainly performed in patients that had a high risk of developing breast and/or ovarian cancer. According to a study conducted by Kuchenbaecker et al. on a large cohort of BRCA1 and BRCA2 carriers, the cumulative risk of (breast cancer) BC by 80 years of age was 72% for BRCA1 carriers and 69% for BRCA2 carriers [2]. However, a significant share of HBOC patients test negative for BRCA mutations, despite positive family or personal history. Multi-gene hereditary cancer panels have been established as detection tools to enhance the detection of a genetic susceptibility for cancer by various gene mutations. The target of screening is to reduce mortality by disease due to the detection of earlier stage cases and to improve patient outcomes.

Following the widespread of next-generation sequencing (NGS) techniques leading to increased accessibility and lower costs, more genes with an increased risk of developing BC have been identified. NGS allows for the sequencing of multiple genes simultaneously and is currently very commonly used in multigene panel hereditary BC risk assessment [3]. Despite these benefits, there are several issues and disadvantages regarding multi-gene testing such as expenses, the identification of low penetrance genes and variants of unknown significance (VUS), and unclear guidelines for the carriers of these “not so common” mutations [3,4].

Besides the BRCA1 and BRCA2 genes, other high penetrance genes such as TP53, CDH1, PALB2, PTEN, and STK11 are involved in the development of BC, as are genes with low risk or insufficient evidence to make any recommendations for patients [5].

The National Comprehensive Cancer Network (NCCN) has also suggested that risk stratification be used in the selection prior to the testing of unaffected and affected women, as it is common knowledge that the lifetime risk of BC in the general population is around 12–13% [5] (Table 1).

Establishing the presence of pathogenic mutations in breast cancer patients is important both in terms of imaging diagnosis and screening, but also in the therapeutic management and subsequent follow-up of these patients [6]. At the same time, correct detection of unaffected but genetically exposed family members should be performed, with appropriate prevention and screening protocols for each mutation type.

Awareness as to the prevalence of these high penetrance mutations in Romanian population is limited and is mainly focused on pathogenic or potentially pathogenic mutations in the BRCA 1 and 2 genes [7].

Our study aimed to identify the high risk pathogenic/likely pathogenic mutations in a cohort of 255 Romanian patients with confirmed BC referred for genetic testing.

## 2. Materials and Methods

Our study is a retrospective analysis of 255 patients diagnosed with BC who presented for oncological examination to the Oncosurg Surgical Oncology Clinic, Cluj-Napoca, between January 2015 and December 2019 and met the National Comprehensive Cancer Network (NCCN) criteria for genetic testing. Overall, 105 patients were excluded from the reference group due to a negative result on genetic testing.

Each patient was approached for genetic testing after being diagnosed with BC. Genomic DNA preparation and sequencing analysis were described in our previous research [8].

Genes were grouped into three risk categories based on penetrance data (Figure 1):High-penetrance breast cancer susceptibility genes: BRCA1, BRCA2, TP53, PALB2, CDH1, STK11, PTENModerate risk genes: ATM, CHEK2, BARD1, RAD51C, RAD51D, NF1Low risk genes: MSH2, MSH6, MLH1, PMS2, EPCAMInsufficient evidence: RAD50, RAD51B, BRIP1, NBN, BLM, FAM175A, MEN1, MRE11A, MUTYH, XRCC2, APC, RET, FANCA

## 3. Results

There were 150 BC patients included in the reference group, in which 177 mutations were identified. Overall, 98 patients had pathogenic mutations and 104 pathogenic variants were diagnosed (six patients had two pathogenic mutations each). Overall, 67 VUS mutations were identified in 52 patients (six patients had two VUS mutations each and nine patients from the pathogenic mutation group also had a VUS mutation).

From the 104 pathogenic mutations identified, 72 mutations were in high-risk genes category and 32 were low- and moderate-risk genes.

We identified 43 pathogenic mutations in the BRCA1 gene in our study group and no VUS mutations. The c.3607C>T variant was the most common mutation diagnosed in the BRCA1 group, followed by c.5266dupC, c.181T>G, and c.3700_3704delGTAAA variants, accounting for almost 72% of all BRCA1 variants (Table 2).

In the BRCA 2 gene category we identified 21 pathogenetic variants and 5 VUS. The c.9371A>T variant was the most common mutation diagnosed in the BRCA2 group, followed by c.8755-1G>A, which accounted for almost 57% of all BRCA2 pathogenic variants (Table 3).

Regarding the TP53 variants, we identified two pathogenic mutations (c.469G>T (2)) and two VUS mutations (c.480G>A, c.847C>T).

In the PALB2 gene, we identified six pathogenetic variants and two VUS mutations (c.2461A>T, c.3122A>C) (Table 4).

In the CDH 1 and STK 11 genes we only identified VUS mutations. In CDH 1 we identified three variants: c.892G>A, c.1840A>G, and c.1297G>A, and in the STK11 gene we identified only the c.1225C>T variant. In the PTEN gene we did not identify any mutations.

There were 21 patients who exhibited overlapping mutations, and their profiles are depicted in Table 5. For high penetrance genes there was just one case of overlapping pathogenic variants for BRCA1 c.3700_3704delGTAAA and BRCA 2 c.9371A>T mutations.

## 4. Discussion

The identification of pathogenic and likely pathogenic gene variants promoters of HBOC is mandatory in the era of patient tailored medicine in order to provide a correct management, prognostic counseling, and follow-up. At the same time, family members of a cancer patient with a HBOC mutation should be offered testing for that precise variant and offered an individualized prevention protocol. Since ancestry and the ethnical and geographical background have been demonstrated to be determinants of one’s genetic profile, it is useful to characterize the specific genetic load of every population. This will allow for the selection of a targeted gene panel for a specific population, reduce costs, and refine therapeutic and surveillance strategies.

### 4.1. BRCA 1

The c.3607C>T variant of BRCA 1 gene was identified in 14 cases of BC patients, all originating from the North-West (8 patients), the West (3 patients), and the Center (3 patients) regions of Romania. This is a nonsense mutation located on exon 10 which produces the amino acid change p.Arg1203Ter, being associated with an increased risk of ovarian and breast cancer [9]. It was previously reported in countries like Greece, Israel, Italy, and Turkey [10], and was mentioned in our previous research as the dominant variant in North-Western Romania [8]. In a study published in 2022 conducted on 490 Romanian patients diagnosed with BC and ovarian cancer (OC), this variant was reported as the most frequent variant associated with OC and the second most frequently associated with BC, accounting for 30 % of BRCA1 mutations in the Romanian population [7].

The c.5266dupC variant of BRCA 1 gene was the second most common mutation identified in BC patients from our study, being identified in 11 cases. This is a pathogenic mutation, located in the coding exon of the BRCA1 gene, which produces a frameshift insertion causing protein truncation and loss-of-function [11]. Although the c.5266dupC pathogenic variant was first described in Ashkenazi Jews, it was later identified in other populations from Europe and South America (Brazil) [11]. It is described as one of the six founder mutations BRCA1 in the Greek population, and is being encountered with increased frequency in Italy [1], Poland [12], Czech Republic [13], Lithuania [14], Belarus [15], Latonia, Russia [16], and the Baltic region [10]. The c.5266dupC variant most likely originated in Northern Europe, specifically Russia or possibly Denmark, between 1500 and 1800 [17].

In our study, the patients identified to have this pathogenic variant originate from North-West region (5 patients), North-East region (3 patients), Western region (2 patients), and the Central region (1 patient) of Romania. A possible explanation for the higher number of patients harboring this variant may be the racial influences of Slavic origin of the population located in the North of Romania due to neighboring with Ukraine, another country with a reportedly high prevalence of this mutation [18] (Figure 2).

In the same study conducted in 2022 on Romanian patients diagnosed with BC and OC, the c.5266dupC variant was reported as the most frequent variant associated with BC and the second most frequently associated with OC [7], with these results suggesting that, together with the c.3607C>T variant, these mutations are the most frequent in the Romanian population.

The BRCA1 c.181T>G variant was the third most common variant identified in our study, along with the c.3700_3704delGTAAA variant, with each being identified in three cases. The BRCA1 c.181T>G variant is a missense mutation that has been reported as having a high frequency in African in countries [19] and Southern Poland [20], and has also been encountered in the Eastern and Baltic regions and in countries such as Bulgaria, Croatia, Romania, and Slovenia [10]. We identified this variant in three patients with BC, with all of them originating from the Nordic region of Romania. This variant has also been reported in Romanian population in a previous study in four cases of BC and OC, being the third most common mutation in Romania [7].

The c.3700_3704delGTAAA pathogenic variant is represented by the deletion of 5 nucleotides in exon 10, causing a frameshift mutation, and has been described as a common or founder mutation in the Czech Republic, Germany, and Poland [20,21]. In the Romanian population, this variant was reported in a previous study in one case of OC [7].

The c.2241dupC variant was identified in two cases in our study, both in patients of Hungarian descendance. This mutation was reported in the Romanian population in one case by two previous studies [22,23].

The c.843_846delCTCA variant was also identified in two cases, with this variant being described in the Romanian population in one case of BC by one of our previous studies [24] and in one case of OC [7].

There were also three pathogenic variants in the BRCA1 gene (c.3187C>T, c.4986 + 6T>C, c.5030_5033delCTAA) which were never described in Romanian population.

### 4.2. BRCA 2

BRCA2 c.9371A>T is the most common pathogenic variant identified in our study, accounting for almost half of the cases. This a single nucleotide variant which causes a non-synonymous transversion from adenine to thymine in exon 25 of the BRCA2 gene, which results in a missense mutation, caused by an amino-acid substitution of asparagine to isoleucine at position p.3124 in the BRCA2 protein [25]. This pathogenic variant was reported previously in German [25], Polish, and Slovenian populations [26]. In the Romanian population, the BRCA2 c.9371A>T variant was reported in seven cases of BC and six cases of OC, and was the most common pathogenic variant described. Our research reports similar results [7]. Out of the nine cases identified in our study, four patients originated from North-Western Romania and five patients originated from Central Romania (Sibiu and Brașov County). It is known that there are large communities of Saxons in these regions, which is a possible underpinning for the increased prevalence of this genetic variant.

BRCA2 c.8755-1G>A was identified in three cases. This variant is also a single nucleotide variant described in Czech, Austrian, and Italian populations [1], and was also reported as the most frequent BRCA2 mutation in Central Europe [27]. This variant was previously reported in the Romanian population by one of our studies [24] and more recently, in 2022, in the case of a patient with BC [7].

Of the three patients detected in our study to carry this mutation, two originated from Central Romania, a mother and daughter of Hungarian descent, diagnosed with breast cancer at the ages of 41 and 71 years, respectively.

We also identified 5 VUS mutations: c.3562A>G (2 cases), c.6607G>T, c.6626T>C, c.3547G>C. From the five patients with BC and BRCA2 VUS mutations, two patients had secondary mutations. The patient with BRCA 2 c.6626T>C variant had a pathogenic mutation in the ATM gene (c.5318delA) and the patient with c.3547G>C had another VUS mutation in the ATM gene (c.9077T>G).

### 4.3. TP53

Li–Fraumeni syndrome is an autosomal dominant condition caused by TP53 germline mutations which predisposes carriers to cancer development. The c.469 G>T variant was associated with the occurrence of hepatocarcinoma, being reported for the first time in literature in 1991 [28].

The c.469G>T variant is located in coding exon 4 of the TP53 gene, and this sequence change replaces Valine with Phenylalanine at codon 157 of the TP53 protein (p.Val157Phe), producing a significant decrease in the structural stability of the DNA-binding domain [29]. This pathogenic variant of TP53 gene was identified by many researchers in patients with breast, hepatocellular, squamous cell carcinoma, and adenocarcinoma [30]. It seems to be associated with HER2 overexpression and presents a potentially higher risk of brain metastases [30].

In our previous study we reported two cases of Li–Fraumeni syndrome [24], both originating from Bacău county (North-East region of Romania), both young at the time of BC diagnosis (33 and 41 years old), with no degree of kinship between them. Both patients were HER 2-positive [8], and one of them presented brain metastases 4 years after the diagnosis. The TP 53 c.480G>A genetic variant was also associated with HER2 positivity and BC. At the four year follow-up after the initial diagnosis of left BC, the patient was confirmed with multifocal contralateral BC. One year later, the patient underwent surgery for an ovarian mass which proved to be ovarian fibroid at the pathology examination. The TP53 c.847C>T variant was reported in the Macedonian population as a low-risk BC allele, but current evaluation on genetic databases classifies this mutation as likely benign. In our study the TP53 c.847C>T variant was associated with the NBN c.511A>G mutation, which is classified as VUS and HER2 +, corresponding to the immunohistochemical phenotype of breast tumors with pathogenic TP53 mutations.

### 4.4. PALB2

The c.93dupA pathogenic variant of PALB2 is a frameshift mutation caused by a duplication of A at nucleotide position 93, resulting in protein truncation. It has been reported in patients with a personal and/or family history of BC [31]. All cases identified in our research were from North-Western and Western Romania.

The PALB2 c.509_510delGA is a frameshift variant which creates a premature stop codon 13 and is expected to result in an absent or disrupted protein product. The frameshift mutation c.509_510delGA has been identified as a recurrent mutation in three independent studies that included patients of German, Polish, Russian, and Belarussian descent [32,33,34].

The c.3549C>G variant changes one nucleotide in exon 13 of the PALB2 gene, creating a premature translation stop signal which results in a nonsense protein. This variant has been associated with an increased risk of familial breast, ovarian, or pancreatic cancer [34,35].

The c.79G>T variant creates a premature translational stop signal in the PALB2 gene, causing an absent or disrupted protein product which was associated with breast and vulvar cancer [36].

### 4.5. CDH1

In the CDH 1 gene we only identified VUS mutations: c.892G>A in a pregnant patient without familial history of BC, c.1840A>G and c.1297G>A, both in patients with a family history of digestive cancers.

The patient with the CDH 1 c.1297G>A variant also exhibited the c.3700_3704delGTAAA pathogenic mutation in the BRCA 1 gene.

### 4.6. STK11

For the STK11 gene we identified only one VUS mutation (c.1225C>T), encountered in two first-degree relatives (mother and maternal aunt), both without BC.

Our research is a first for the Romanian population in highlighting the genetic heritage for HBOC in our region, other than BRCA 1/2 genes. This type of reporting should be extended for the Romanian population to allow for the design of an individualized management protocol. In fact, the final goal of global testing would be to obtain a genetic map of HBOC risk. This would allow for the stratification of cancer likelihood of individuals based on age, ancestry, and geographic region. Our study could be the prototype or an appendix of the worldwide HBOC genome.

The early detection of cancer risk allows for the implementation of specific prevention protocols and even targeted drug therapies, such as those based on PARP (Poly (ADP-ribose) polymerase) inhibitors [37]. Further genetic analysis by whole exome sequencing and targeted analysis might be readily available and recommended in the future, either as a first or second line approach for the genetic testing of HBOC patients [38,39,40].

## 5. Conclusions

This is one of the first studies to evaluate the contribution of high-risk germline mutations to BC development in the Romanian population, as most of the research was carried out by researchers that were only focused on the BRCA 1 and 2 genes.

All the BRCA1 mutations identified were pathogenic variants according to the ClinVar database. The most frequent BRCA1 gene mutation detected in our study population was c.3607C>T, similar to other reports concerning the Romanian population. The c.5266dupC, c.181T>G variants, reported with increased frequency in the population from North and North-Western region of Romania, may be due the Slavic racial influences of Ukraine in the population from Northern Romania.

The BRCA2 c.9371A>T variant was the most frequent pathogenic variant described in our group, similar to other studies reported on the Romanian population. This could be due to the heritage from Saxons communities in Central and North-West Romania. The third most frequently encountered gene with pathogenic variants was PALB2, and the most frequent mutation was c.93dupA in three of the six pathogenic variant cases. Regarding mutations of the TP 53 gene, the pathogenic variant c.469G>T was identified in two cases, which were both associated with HER 2 overexpression. We identified only VUS-type variants in both STK11 and CDH 1 genes, and no mutations in the PTEN gene.

This study provides a cartograph of high-penetrance breast cancer susceptibility genes in Romanian patients, highlighting potential racial influences from neighboring populations.

## Figures and Tables

**Figure 1 cancers-15-01895-f001:**
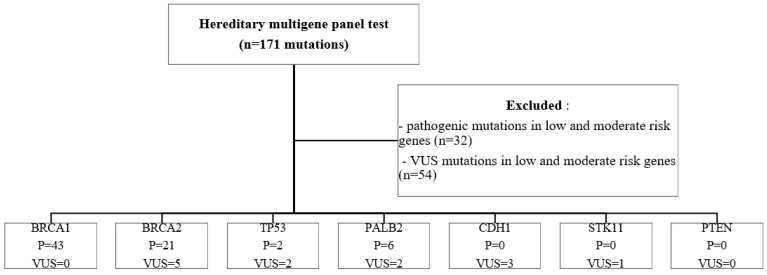
Study population—inclusion and exclusion criteria. P—pathogenic; VUS—variants of unknown significance.

**Figure 2 cancers-15-01895-f002:**
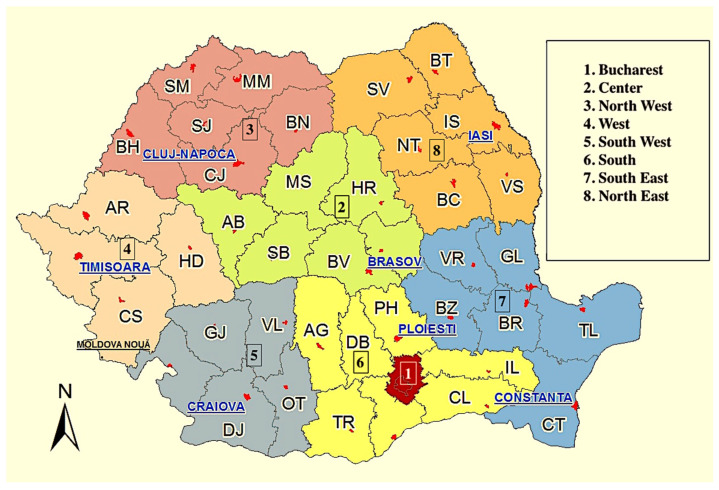
Geographical regions of Romania (adapted figure)**.**
http://www.cniptmoldovanoua.ro/harta-regiuni-romania/ accessed on the 14 March 2023.

**Table 1 cancers-15-01895-t001:** Absolute lifetime risk for BC according to NCCN [5].

>60%	41–60%	15–40%	<15%
BRCA1	CDH 1	ATM	MLH1
BRCA2	PALB2	BARD1	MSH2
TP53	PTEN	CHEK2	MSH6
	STK11	NF1	PMS2
		RAD51C	EPCAM
		RAD51D	

**Table 2 cancers-15-01895-t002:** Pathogenic BRCA 1 variants identified in the study group patients.

HGVS Mutation	Cases	Percentage	Variant	Type
c.3607C>T	14	32.55	Pathogenic	Nonsense
c.5266dupC	11	25.58	Pathogenic	Frameshift duplication
c.181T>G	3	6.97	Pathogenic	Missense
c.3700_3704delGTAAA	3	6.97	Pathogenic	Frameshift deletion
c.2241dupC	2	4.65	Pathogenic	Frameshift duplication
c.843_846delCTCA	2	4.65	Pathogenic	Frameshift deletion
c.135-2A>G	1	2.33	Pathogenic	Frameshift–splice acceptor
c.4035delA	1	2.33	Pathogenic	Frameshift deletion
c.1789G>A	1	2.33	Pathogenic	Nonsense
c.737delT	1	2.33	Pathogenic	Frameshift deletion
c.3187C>T	1	2.33	Pathogenic	Single nucleotide variant
c.4986 + 6T>C	1	2.33	Pathogenic	Single nucleotide variant
c.212 + 1G>T	1	2.33	Pathogenic	Single nucleotide variant
c.5030_5033delCTAA	1	2.33	Pathogenic	Frameshift deletion

**Table 3 cancers-15-01895-t003:** Pathogenic BRCA 2 variants identified in the study group patients.

HGVS Mutation	Cases	Percentage	Variant	Type
c.9371A>T	9	42.85	Pathogenic	Missense
c.8755-1G>A	3	14.28	Pathogenic	Frameshift–splice acceptor
c.1528G>T	1	4.76	Pathogenic	Nonsense
c.9253delA	1	4.76	Pathogenic	Frameshift deletion
c.7007G>C	1	4.76	Pathogenic	Missense
c.8695C>T	1	4.76	Pathogenic	Nonsense
c.7209_7212delCAAAinsGG	1	4.76	Pathogenic	Frameshift deletion
c.6557C>A	1	4.76	Pathogenic	Single nucleotide variant
c.793 + 1G>A	1	4.76	Pathogenic	Single nucleotide variant
c.3462delC	1	4.76	Pathogenic	Frameshift deletion
c.8655dupA pat	1	4.76	Pathogenic	Frameshift duplication

**Table 4 cancers-15-01895-t004:** Pathogenic PALB2 variants.

HGVS Mutation	Cases	Percentage	Variant	Type
c.93dupA	3	50%	Pathogenic	Frameshift duplication
c.509_510delGA	1	16.6%	Pathogenic	Frameshift deletion
c.3549C>G	1	16.6%	Pathogenic	Nonsense
c.79G>T	1	16.6%	Pathogenic	Nonsense

**Table 5 cancers-15-01895-t005:** Overlapping mutations registry of the study group patients.

No	Pathogenic Variant	Pathogenic Variant	VUS	VUS
1	BRCA 1 c.843_846delCTCA	CHEK2 c.470T>C	-	-
2	RAD51C c.905-2A>G	MUTYH c.536A>G	-	-
3	ATM c.5318delA	MUTYH c.721C>T	-	-
4	ATM c.7630-2A>C	CHEK2 c.444\ + 1G>A	-	-
5	BRCA1 c.3700_3704delGTAAA	BRCA 2 c.9371A>T	-	-
6	CHEK 2 1283C>T	BLM c.1642C>T	-	-
7	BRCA1 c.135-2A>G	-	MUTYH c.158-3C>T	
8	BRCA1 c.2241dupC	-	MEN1 c.777G>A	-
9	BRCA 1 c.3700_3704delGTAAA	-	CDH1 c.1297G>A	-
10	BRCA 1 c.3700_3704delGTAAA	-	ATM c.2735G>A	-
11	BRCA 1 c.5266dupC	-	FANCM c.1576C>G	-
12	BRCA 2 c.9371A>T	-	ATM c.2735G>A	-
13	CHEK2 c.1232G>A	-	RAD50 c900G>A	-
14	NBN c.657_661delACAAA	-	MRE11A c.1091G>A	-
15	BRCA2 c.9371A>T	-	RAD51B c.976C>G	-
16	-	-	ATM c.9077T>G	BRCA2 c.3547G>C
17	-	-	BARD 1 c.1915T>C	RAD51C c.790G>A
18	-	-	NBN c.511A>G	TP53 c.847C>T
19	-	-	MSH6 c.2189A>G	RAD51C c.1063G>A
20	-	-	RAD50 c.1663A>G	RAD51C c.790G>A
21	-	-	ATM c.1444A>C	FANCA c.2715A>G

## Data Availability

All data is available from the corresponding author upon request.

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
