# Peer review of "Spectrum of High-Risk Mutations among Breast Cancer Patients Referred for Multigene Panel Testing in a Romanian Population"

_cancers, 2023, doi:10.3390/cancers15061895_

Round 1
Reviewer 1 Report
The authors uncover the spectrum of high-risk mutations among breast cancer patients referred for multigene panel testing in a Romanian population.
The manuscript is of interest, nevertheless, the next points should be addressed
a) Authors report a lack of novelty and lack of rationale for why they came up with this research: highlight what this manuscript could add. How did the authors come up with this information? Describe the information that isn't exactly available that motivated the authors to come up with it.
The authors highlight current research in response to the current caveats. Not addressing them in the background and future directions will be necessary if they do not.
b) High-resolution figuresare required for the figures: they should be state-of-the-art.
c) During the design and enrollment of clinical trials, the authors should consider genomic-directed stratifications, also in light of their findings. As a whole, I think that well-designed clinical trials are required to test more precision and individualized approaches, which is a challenge. I would like to hear how they envision this being achieved. It should be highlighted as a limitation if this is beyond the scope of the manuscript
d) Discuss how the research contributes to research in progress by highlighting the new information the review provides
e) Did the author correct for biases and confounding factor potentially affecting the survival and, after having assumed an hazard's proportionality, did they perform the correct cox multivariable model?
f)There is many potential biais in this study that have to be discussed in the manuscript as the authors performed many statistical analysis that could conduct to a false correlation between those factors. It should have been interesting to confirm this hypothesis on a external validation cohort.
g)Another question is the accuracy of clinical data about the genomic presence as a risk signature in the data base and the possible interaction between the higher rate of comorbidities in the given population and the occurence of different onset of breast cancer. are these data really reliable?
h) Is this data set already audited on this topic?
i)as a mix of bioinformatic study and validation, an additional external cohort should be included to validate their findings or this should be highlighted as limitaion
l)No validation work was performed in additional clinical samples with/without the given mutations. At least, they can perform some in-vitro assay to confirm their findings in cell lines with/without the given signature or alternative model.
m) The writing of whole paper was too simple. The abstract, method, results, and discussion part.
n) this reviewer misses insights regarding novel aspects that should be stressed. As is now well known, tumors grow and evolve through a constant crosstalk with the surrounding microenvironment, and emerging evidence indicates that angiogenesis and immunosuppression frequently occur simultaneously in response to this crosstalk. Accordingly, strategies combining anti-angiogenic therapy and immunotherapy seem to have the potential to tip the balance of the tumor microenvironment and improve treatment response. Please refer to PMID: 34440380 and expand.
Minor
1. do the author propose an exomic sequencing procedure or a targeted analysis?
2. clarify the methodology used to distinguish somatic mutations from germline mutations
3. the method that can be used for prospective validation studies could be that of DNA extraction and next generation sequencing (i.e. Illumina MiSeq), with various custom panels (I don't know, there are some of an immuno-haematological nature). The coding regions and intron-exon junction that can be analyzed could in this case should be focused on how many genes? The most authoritative literature about the panels identifyied, and the regions chosen would be analyzed with coverage >20X: >95%? Can the author give their idea?
4. a second option would be that of an exomic sequencing procedure (can you add some specifics about it?)
Author Response
Reviewer 1
Esteemed Reviewer,
We really appreciate your letter and the comments with suggestions concerning our manuscript. We made a point-by-point revision of the manuscript according to your observations and all changes are highlighted in blue in the text. We hope you find our revision acceptable.
Reviewer: Authors report a lack of novelty and lack of rationale for why they came up with this research: highlight what this manuscript could add. How did the authors come up with this information? Describe the information that isn't exactly available that motivated the authors to come up with it.The authors highlight current research in response to the current caveats. Not addressing them in the background and future directions will be necessary if they do not.
Authors: Thank you for your comment. As stated in the introduction (lines 84-85) awareness as to the prevalence of high penetrance mutations in Romanian population is limited and mainly focused on pathogenic or probably pathogenic mutations in the BRCA 1 and 2 genes.
Establishing the presence of pathogenic mutations in breast cancer patients is important both in terms of imaging diagnosis and screening, but also in the therapeutic management and subsequent follow-up of patients. The detection of unaffected but genetically exposed family members should be performed, with appropriate prevention and screening protocols for each mutation type.
Reviewer: High-resolution figuresare required for the figures: they should be state-of-the-art.
Authors: The article provides 5 tables and one scheme designed by us and a map of Romania useful to depict the geographical provenance of our study population. Which is not state-of the art? We would very much like to comply with your requirements.
Reviewer: During the design and enrollment of clinical trials, the authors should consider genomic-directed stratifications, also in light of their findings. As a whole, I think that well-designed clinical trials are required to test more precision and individualized approaches, which is a challenge. I would like to hear how they envision this being achieved. It should be highlighted as a limitation if this is beyond the scope of the manuscript.
Authors: We thank you for your comment and truly appreciate your point. The high-end result of genetic testing for HBOC would be to the construct the global genetic map based on ancestry and geographical regions. This would allow the stratification of risk according to age, inheritance as well as other risk factors. Ours is just a tiny piece of the immense puzzle. We have appended our discussions sections accordingly.
Reviewer: Discuss how the research contributes to research in progress by highlighting the new information the review provides.
Authors: Thank you very much for your advice. We have emphasized the role of our results in the discussion section, lines 288-291. We know this is not innovative research, we are not proposing the implementation of a new technique, or a screening protocole. But the we support the hypothesis that characterization of the genetic background of a certain population may allow for a targeted smaller scale and lower cost genetic screening recommendation for HBOC. (doi: 10.1056/NEJMoa2005936, doi:10.1155/2020/6384190, doi: 10.3389/fgene.2023.1060504)
Reviewer: Did the author correct for biases and confounding factor potentially affecting the survival and, after having assumed an hazard's proportionality, did they perform the correct cox multivariable model?
Authors: I hope you will not consider us thick-headed, but we do not fully understand your question. We do not take into discussion survival rates, therefore there is no issue of confounding, nor of association to predictor variables.
Reviewer: There is many potential biais in this study that have to be discussed in the manuscript as the authors performed many statistical analysis that could conduct to a false correlation between those factors. It should have been interesting to confirm this hypothesis on a external validation cohort.
Authors: Again, we do not understand. All breast cancer patients referred to our centre were screened. It is a report of screening results. The study plot is quite significant in size. Which kind of bias are you referring to?
Reviewer: Another question is the accuracy of clinical data about the genomic presence as a risk signature in the data base and the possible interaction between the higher rate of comorbidities in the given population and the occurence of different onset of breast cancer. are these data really reliable?
Authors: The genetic underpinnings of HBOC could be involved in the onset of other comorbidities, if this is your point. But this does not make genes less or more responsible for trigerring cancer. At the same time, we did not discuss age at BC detection. This is beyond the scope of our research.
Reviewer: Is this data set already audited on this topic?
Authors: We do not understand your question. Do you wish to see the study/gene results database?
Reviewer: As a mix of bioinformatic study and validation, an additional external cohort should be included to validate their findings or this should be highlighted as limitaion
Authors: This is a descriptive study, not a prognostic model. Why would it need validation?
Reviewer: No validation work was performed in additional clinical samples with/without the given mutations. At least, they can perform some in-vitro assay to confirm their findings in cell lines with/without the given signature or alternative model.
Authors: This is a descriptive study, not a prognostic model. Why would it need validation?
Reviewer: The writing of whole paper was too simple. The abstract, method, results, and discussion part.
Authors: We have appended the manuscript in several parts, all highlighted in blue. We appreciate your comment, it is indeed not very literary. But it was our aim to keep it simple and just present data. Should you suggest we emphasize more, please help us by suggesting where to insist.
Reviewer: this reviewer misses insights regarding novel aspects that should be stressed. As is now well known, tumors grow and evolve through a constant crosstalk with the surrounding microenvironment, and emerging evidence indicates that angiogenesis and immunosuppression frequently occur simultaneously in response to this crosstalk. Accordingly, strategies combining anti-angiogenic therapy and immunotherapy seem to have the potential to tip the balance of the tumor microenvironment and improve treatment response. Please refer to PMID: 34440380 and expand.
Authors: I see your point, this type of research is highly innovative and will probably give the future direction for clinical practice, but not for a while. For the time being, we as clinicians aim to offer the best approach for our patients and their families, and it is this the motivation behind our whole endeavor. Maybe you will find it too bold or too sincere of me, but such fundamental research is sometimes even difficult to integrate for the common MD of a clinical specialty.
Minor comments
Reviewer: Do the author propose an exomic sequencing procedure or a targeted analysis?
Authors: Thank you for your comment. As far as we know, there are no current guidelines to suggest such costly investigations, nor recent research in support of this recommendation. As a personal opinion, perhaps the near future holds more for targeted analysis that WES.
Reviewer: Clarify the methodology used to distinguish somatic mutations from germline mutations.
Authors: I appreciate your clarification. All data was obtained by blood testing and NGS, we did not look for somatic mutations in these patients.
Reviewer: The method that can be used for prospective validation studies could be that of DNA extraction and next generation sequencing (i.e. Illumina MiSeq), with various custom panels (I don't know, there are some of an immuno-haematological nature). The coding regions and intron-exon junction that can be analyzed could in this case should be focused on how many genes? The most authoritative literature about the panels identifyied, and the regions chosen would be analyzed with coverage >20X: >95%? Can the author give their idea?
Authors: I apologize once again for my reply. The reply is probably also suitable for your last question. We are clinicians, that is mainly gynecologists and surgeons. This question is too technical for us and far beyond the aim of our study.
Reviewer: A second option would be that of an exomic sequencing procedure (can you add some specifics about it?)
Authors: WES is very expensive and results are difficult to interpret. Perhaps the next couple of years will bring more clarification to this potential next step.

Reviewer 2 Report
Comments:
1. Can authors provide the info that how many overlapped mutations in one patient? For example, on Table 2, the first 2 mutations have more than 10 cases. How many patients have both mutations?
2. Any data on p53 mutations?
3. Any relationship between Figure 1 and mutations shown in the current manuscript?
Author Response
Reviewer 2
Thank you for your comments following the revision of our work. We have responded to all suggestions individually and all changes are highlighted in green in the manuscript.
- Can authors provide the info that how many overlapped mutations in one patient? For example, on Table 2, the first 2 mutations have more than 10 cases. How many patients have both mutations?
We appreciate your comment and below you can find the list of all overlapping mutations. More so, even, we have prepared a table and added it to the results section to make things clear for all readers.
The following 6 patients had 2 pathogenic mutations each:
- L 43 years - BRCA 1 c.843_846delCTCA (pathogenic) and CHEK2 c.470T>C (pathogenic)
- M. 41 years - RAD51C c.905-2A>G (pathogenic) and MUTYH c.536A>G (pathogenic)
- D 43 years - ATM c.5318delA (pathogenic) and MUTYH c.721C>T (pathogenic)
- S. 43 years - ATM c.7630-2A>C and CHEK2 c.444\+1G>A
- M 70 years - BRCA 2 c.9371A>T and BRCA1 c.3700_3704delGTAAA
- L 41 years - CHEK 2 1283C>T and BLM c.1642C>T (pathogenic)
9 patients from pathogenic mutation group had also a VUS mutation:
- L. 45 years – BRCA1 c.135-2A>G (pathogenic) and MUTYH c.158-3C>T (VUS)
- N. 40 years - BRCA1 c.2241dupC (pathogenic) and MEN1 c.777G>A (VUS)
- I. 32 years - BRCA 1 c.3700_3704delGTAAA (pathogenic) and CDH1 c.1297G>A (VUS)
- L . 59 years - BRCA 1 c.3700_3704delGTAAA and ATM c.2735G>A (VUS)
- M 41 years - BRCA 1 c.5266dupC and FANCM c.1576C>G (VUS )
- I. 36 years - BRCA 2 c.9371A>T (pathogenic) and ATM c.2735G>A (VUS)
- C 45 years - CHEK2 c.1232G>A (pathogenic) and RAD50 c900G>A (VUS)
- C 53 years - NBN c.657_661delACAAA (pathogenic) and MRE11A c.1091G>A (VUS)
- I. - BRCA2 c.9371A>T and RAD51B c.976C>G (VUS)
The following 6 patients had 2 VUS mutations each and
- D. 42 years - ATM c.9077T>G and BRCA2 c.3547G>C
- T 45 years - BARD 1 c.1915T>C and RAD51C c.790G>A
- O. - NBN c.511A>G And TP53 c.847C>T
- D. 36 years MSH 6 c.2189A>G and RAD51C c.1063G>A
- A. 41 years - RAD 50 c.1663A>G and RAD51C c.790G>A
- S 35 years - ATM c.1444A>C and FANCA c.2715A>G
Table . Overlapping mutations registry of the study group patients.
|
No |
Pathogenic variant |
Pathogenic variant |
VUS |
VUS |
|
1 |
BRCA 1 c.843_846delCTCA |
CHEK2 c.470T>C |
- |
- |
|
2 |
RAD51C c.905-2A>G |
MUTYH c.536A>G |
- |
- |
|
3 |
ATM c.5318delA |
MUTYH c.721C>T |
- |
- |
|
4 |
ATM c.7630-2A>C |
CHEK2 c.444\+1G>A |
- |
- |
|
5 |
BRCA1 c.3700_3704delGTAAA |
BRCA 2 c.9371A>T |
- |
- |
|
6 |
CHEK 2 1283C>T |
BLM c.1642C>T |
- |
- |
|
7 |
BRCA1 c.135-2A>G |
- |
MUTYH c.158-3C>T |
|
|
8 |
BRCA1 c.2241dupC |
- |
MEN1 c.777G>A |
- |
|
9 |
BRCA 1 c.3700_3704delGTAAA |
- |
CDH1 c.1297G>A |
- |
|
10 |
BRCA 1 c.3700_3704delGTAAA |
- |
ATM c.2735G>A |
- |
|
11 |
BRCA 1 c.5266dupC |
- |
FANCM c.1576C>G |
- |
|
12 |
BRCA 2 c.9371A>T |
- |
ATM c.2735G>A |
- |
|
13 |
CHEK2 c.1232G>A |
- |
RAD50 c900G>A |
- |
|
14 |
NBN c.657_661delACAAA |
- |
MRE11A c.1091G>A |
- |
|
15 |
BRCA2 c.9371A>T |
- |
RAD51B c.976C>G |
- |
|
16 |
- |
- |
ATM c.9077T>G
|
BRCA2 c.3547G>C |
|
17 |
- |
- |
BARD 1 c.1915T>C |
RAD51C c.790G>A |
|
18 |
- |
- |
NBN c.511A>G |
TP53 c.847C>T |
|
19 |
- |
- |
MSH6 c.2189A>G |
RAD51C c.1063G>A |
|
20 |
- |
- |
RAD50 c.1663A>G |
RAD51C c.790G>A |
|
21 |
- |
- |
ATM c.1444A>C |
FANCA c.2715A>G |
In the case of high penetrance genes there was just one case of overlapping pathogenic variants for BRCA1 c.3700_3704delGTAAA and BRCA 2 c.9371A>T mutations.
- Any data on p53 mutations?
Thank you for your valuable questions. We had already included a comment in the manuscript at lines 227-230 and 270: Both patients with TP 53 c.469 G>T pathogenic variant were from the same region of Romania, with no degree of kinship. Both were Her 2 + which is common for these mutations and presented with brain metastases.
We have included the following points to expand on the aspect.
The TP 53 c.480G>A genetic variant was also associated with HER2 + BC. At the 4 years follow-up after the initial diagnosis of left BC, the patient was confirmed with multifocal contralateral BC. One year later, the patient underwent surgery for an ovarian mass which proved to be ovarian fibroma at the pathology examination.
The TP53 c.847C>T variant was reported in Macedonian population as a low-risk BC allele, but current evaluation on genetic databases classify this mutation as likely benign. In our study the TP53 c.847C>T variant was associated with the NBN c.511A>G mutation which is classified as VUS and was HER2+, corresponding to the immunohistochemical phenotype of breast tumors with pathogenic TP53 mutations.
- Any relationship between Figure 1 and mutations shown in the current manuscript?
Figure 1 represents the distribution of patients with pathogenic mutations in high penetrance genes, related to the geographical regions in Romania.

Round 2
Reviewer 1 Report
I am satisfied with the rebuttal provided.
Reviewer 2 Report
no more comments